# The Trajectory of Anthropomorphism and Pro-Environmental Behavior: A Serial Mediation Model

**DOI:** 10.3390/ijerph20032393

**Published:** 2023-01-29

**Authors:** Yiping Yang, Le Sun, Buxin Han, Pingping Liu

**Affiliations:** 1CAS Key Laboratory of Mental Health, Institute of Psychology, Beijing 100101, China; 2Department of Psychology, University of Chinese Academy of Sciences, 19A Yuquan Road, Beijing 100049, China

**Keywords:** anthropomorphism of nature, connectedness to nature, environmental guilt, pro-environmental behavior, age differences, mediating effect

## Abstract

Anthropomorphism of nature can promote pro-environmental behavior (PEB). However, its underlying mechanism and these age-related changes are unclear. We propose that connectedness to nature and environmental guilt mediate the relationship between anthropomorphism of nature and PEB. The present study tests the hypotheses based on a cross-sectional sample of 1364 residents aged 15–76 years, using structural equation modeling. We found that: (1) environmental guilt decreases, but PEB increases, with age; (2) anthropomorphism of nature decreases in early adulthood and increases in old age; (3) connectedness to nature decreases in mid–late adolescence and increases in early adulthood. Connectedness to nature and environmental guilt have a serial mediating effect in the relationship between anthropomorphism of nature and PEB, with cross-age stability. These findings contribute to enriching the understanding of PEB from the human and nature perspective, and enhancing anthropomorphism of nature that could promote PEB in residents at different ages, through connectedness to nature and environmental guilt.

## 1. Introduction

People live in nature. However, human behavior is a fundamental cause of environment deterioration. An important reason is anthropocentrism, which focuses on human development regardless of natural sustainability [1]. Humans perceive nature as an object to plunder and conquer, and make it fragments [2,3]. For example, due to the misuse of natural resources, global greenhouse gas emissions set a new record, with an average temperature increase of 1.25 °C, and a glacier loss of 31% more snow and ice in 2021 [4]. Thus, in order to halt the degradation of nature, it is urgent to encourage individuals to engage in environmentally-friendly lifestyles, and implement pro-environmental behavior (PEB), which refers to all behaviors that protect our environment and minimize the negative impact on the environment [5,6,7,8,9].

Since the 1960s, factors influencing PEB have been investigated, such as past experience, climate change beliefs, environmental knowledge, commitment, and environmental attitudes [10,11,12,13,14]. However, surprisingly little attention has been given to the domain of nature, such as anthropomorphism and nature connection [15]. Harmonious relationship between nature and humans is crucial for sustainable development [1,3,6,16]. Scholars have attempted to develop an integrated, sustainable system between nature and humans, and critique the human-centric perspective [3]. By emphasizing the similarity between nature and humans (e.g., the Earth has a fever), anthropomorphism of nature can bring people closer to nature, and establish a sense of oneness with nature [17,18,19]. Furthermore, research has indicated that anthropomorphism of nature is positively associated with pro-environmental attitudes or behaviors [20]. Anthropomorphism could lead to more PEBs (e.g., people are willing to pay for conservation) [21,22]. However, the underlying mechanisms remain unclear [23]. This study aims to address this question.

There are two potential mediators of the relationship between anthropomorphism of nature and PEB. First, it is possible that the anthropomorphism of nature influences PEB through connectedness to nature, which refers to an emotional connection with nature [24,25]. Second, environmental guilt, or the guilt experienced by people for harming nature, may be another mediator between anthropomorphism of nature and PEB [26]. However, previous studies have only considered one of these mediators and lacked extensive discussion.

To our knowledge, there have been no studies exploring connectedness to nature and environmental guilt together for PEB. Do these two variables influence anthropomorphism or does one take precedence? Additionally, we know very little about the relative strength of each mediator in the relationship between anthropomorphism and pro-environmental behaviors.

It is noteworthy that age is also a crucial factor influencing PEB [10,27,28]. Typically, as people age, they become more prone to engaging in PEB, such as green consumption, energy saving, and recycling [28,29,30]. However, most studies on the effects of anthropomorphism on PEB focused on people in early adulthood [18,22,24,26]. Despite the important role of anthropomorphism in PEB, little is known about its developmental course across different age groups. The unresolved question is whether there are age-related differences in the effects of anthropomorphism of nature on PEB.

Therefore, based on the environmental view of unity between nature and humans, the present study aims to explore the mechanisms by which anthropomorphism of nature promotes PEB from a holistic view, and its age-related differences. In particular, the contributions of this study are as follows. First, we explore the impact of anthropomorphism of nature on PEB through connectedness to nature and environmental guilt. This can clarify its path, and compensate for the deficiency of existing literature in explaining its internal mechanism. Second, we discuss age differences in the internal mechanism, as well as these trajectories of variables, contributing to the existing studies on age-related changes. Third, from theoretical and practical aspects, we provide suggestions to policymakers that it is crucial for people of different ages to effectively implement green anthropomorphic strategy in the context of unity between nature and human.

## 2. Conceptual Framework and Hypothetical Development

### 2.1. Anthropomorphism of Nature and Pro-Environmental Behavior

Anthropomorphism of nature refers to the imbuing of imagined or real behavior in nature with human-like characteristics, intentions, emotions, and motivations [17,31]. Anthropomorphism of nature is a broad and inclusive concept that emphasizes the perception of human characteristics of nature. For example, people describe nature as Mr. Nature or Mother Earth, or incorporate human elements, such as eyes and mouths, in their nature-related designs [18,24]. It is worth noting that anthropomorphism of nature is not only about assigning human behavior or appearance to nature. It also involves unique human mental abilities, such as consciousness, free will, or social mindfulness [32].

Anthropomorphism could elicit cognitive responses similar to real human contact, and this makes it possible to apply moral judgments to nonhuman agents [17,18]. Through anthropomorphism, nature is personified and could be perceived as having a human-like mind. This means that nature can be endowed with the ability to experience consciousness and emotions [32]. Anthropomorphism could further moralize nature and make it a moral agent of concern, so that nature would be respected and protected morally [22,23]. Individuals are unlikely to harm other human beings, because harm is considered immoral. Thus, they would also be unlikely to harm nature, which has thoughts and feelings [33,34,35,36]. For example, a poster with anthropomorphic content could motivate people to use environmentally friendly products and support the environmental movements [18]. Likewise, anthropomorphism of nature was positively correlated with observed green donations [21]. Thus, the first hypothesis of this study is as follows:

**Hypothesis** **(H1).***Anthropomorphism of nature positively affects PEB*.

However, the role of anthropomorphism in promoting PEB is conditional [20]. Anthropomorphism of nature tends to significantly influence PEB through the mediation of other factors [23]. One of the critical factors may be the connectedness to nature.

### 2.2. Connectedness to Nature

Since ancient times, people have had the concept of unity between nature and humans, and have focused on this relationship [37]. Connectedness to nature refers to how people emotionally belong to nature, a subjective sense of oneness with nature, and self-definition as part of the natural world [38,39]. Connectedness to nature describes the interconnectedness of humans and nature [38]. Researchers suggest that it is similar to interpersonal connection [18,40]. The need for social connection is one of the basic motivations of human beings, since it is crucial for survival and health [41,42]. Mayer et al. [40] argue that people can satisfy their needs for social connection through connectedness to nature. Moreover, connectedness to nature can be enhanced by anthropomorphizing nature [18,24]. According to the three-factor theory of anthropomorphism, by emphasizing the similarity between humans and nonhuman agents, anthropomorphism enables people to establish connections with nonhuman agents and thus satisfy their needs for social connection [17]. When nature is anthropomorphized, people will realize its similarity to humans and develop similar connections with nature, ultimately satisfying the need for social connection. For instance, “Mother Nature” could guide people to integrate nature into the self and enhance the human-nature connection [24], and people in the anthropomorphic condition have a stronger emotional connection with nature [18].

Connectedness to nature is the emotional bond that people have with nature. People who have a stronger sense of connection with nature will consider themselves belonging to nature and perceive nature’s well-being as their own [37]. Therefore, people will care more about natural sustainability [37]. When the self is highly integrated with nature, people will make a commitment to protect nature. This commitment represents a willingness to maintain a close relationship with nature, and invest resources and time in maintaining that relationship, so that people will be more willing to act in the benefits of nature [38]. A recent meta-analysis indicated that the relationship between connectedness to nature and PEB was positive, significant, and moderately sized, and this relationship was held across home location, gender, and age [43]. Thus, we assume that connectedness to nature could promote PEBs [6]. People who have a stronger sense of connection to the natural world, or feel intimate with nature, will be more likely to protect it.

In short, anthropomorphism of nature promotes connectedness to nature, which significantly and positively predicts PEB. Thus, we hypothesize that connectedness to nature mediates the relationship between anthropomorphism of nature and PEB. Therefore, we posit the second hypothesis:

**Hypothesis** **(H2).***Connectedness to nature has a positive mediating effect between anthropomorphism of nature and PEB*.

### 2.3. Environmental Guilt

Environmental guilt may be another mediator between anthropomorphism of nature and PEB. Environmental guilt refers to one’s collective guilt towards nature [26,44]. People can experience this negative emotion when their in-group members behave immorally or harm others [45]. That is, if people perceive that their group members engage in environmentally-harmful behaviors, those who are normally concerned about environmental protection can also experience environmental guilt [44].

However, why do people feel guilty about environmental degradation? One reason is that anthropomorphism makes individuals attribute nature in a similar manner to human beings, such as having minds and emotions [17,31]. It is well known that guilt is usually evoked during interpersonal interactions. By anthropomorphizing nature, people can perceive nature as a member similar to humans [19]. In this way, individuals can treat nature in a way that is similar to how they treat other people. As a result, people may feel guilty about harming nature. A cross-cultural study showed that anthropomorphism of nature could induce environmental guilt [26]. Therefore, it is possible for people to experience guilt when they believe that humans harm nature.

Although guilt is an unpleasant emotion associated with the experience of anxiety or distress, it is considered as “adaptive” and is closely related to pro-social behavior [46]. According to the interpersonal guilt theory, individuals adopt compensatory behaviors to reduce guilt experiences and rebuild good relationships [47]. Similarly, environmental guilt is experienced when people perceive that in-group members harm the natural environment. In order to reduce negative feelings and compensate for nature, engaging in PEB is one of the most common responses [44]. For example, people experience more guilt if they believe that humans cause global warming, and consequently show more support for pro-environmental policies [45]. Harth et al. [48] also found that individuals’ guilt for air pollution positively predicted their intention to repair the damage. Therefore, when an in-group member causes environmental problems, individuals develop environmental guilt toward nature and engage in more PEBs [26].

In summary, anthropomorphism of nature positively affects environmental guilt. Environmental guilt is an essential factor that influences PEB. Thus, environmental guilt may be a mediator between anthropomorphism of nature and PEB. Thus, we propose the following hypothesis:

**Hypothesis** **(H3).***Environmental guilt has a positive mediating effect in the relationship between anthropomorphism of nature and PEB*.

### 2.4. Serial Mediating Effect

As described above, we claim that anthropomorphism of nature can positively influence PEB through the separate mediating effects of connectedness to nature and environmental guilt. However, several studies show that the link between anthropomorphism of nature and environmental guilt is weak, indicating that there may be other strong predictive factors of environmental guilt [23,26]. Thus, we speculate that connectedness to nature may positively predict the level of environmental guilt. Guilt is derived from interpersonal relationships, and it contributes to the maintenance of good interpersonal relationships [46,48,49,50]. The experience of guilt will be stronger and more influential in close relationships than that in distant ones [47]. Interpersonal characteristics can be applied to nature–human relationships [26,37], since connectedness to nature represents the relationship between humans and nature [38]. Therefore, the present study links connectedness to nature and environmental guilt. We speculate that the more connected individuals are to nature, the more likely they are to feel guilty about the environmental degradation caused by humans.

Thus, anthropomorphism of nature may influence PEB through a new path. We hypothesize that anthropomorphism of nature causes individuals to attribute human traits to nature, bringing people closer to nature and creating an emotional connection with nature. Based on the intimate connection with nature, people are more likely to experience guilt for human-caused environmental problems. Finally, individuals will perform PEBs to improve their relationship with nature and reduce the negative experience of guilt. In conclusion, the present study aims to explore the mechanisms by which anthropomorphism of nature promotes PEB. Thus, we propose the following hypotheses:

**Hypothesis** **(H4a).**
*Connectedness to nature is strongly associated with environmental guilt.*


**Hypothesis** **(H4b).**
*Anthropomorphism of nature impacts PEB through the serial mediating effect of connectedness to nature and environmental guilt.*


The hypothesized research model is shown in Figure 1.

### 2.5. Age-Related Differences

Many complex human traits vary across the lifespan, such as cognition, emotion, or socialization [51,52,53,54,55,56,57,58,59,60]. Thus, anthropomorphism of nature, connectedness to nature, environmental guilt, and relationships among these variables may also fluctuate with age. However, to our knowledge, no studies have holistically reported the life-span trajectories of anthropomorphism of nature, connectedness to nature, environmental guilt, and PEB, especially for residents aged from 15 to 76 years. To address this gap, we focus on individuals in mid–late adolescence, early adulthood, middle adulthood, and old age.

There are different developmental characteristics among individuals in mid–late adolescence, early adulthood, middle adulthood, and old age. Mid–late adolescence (15–20 years old) is a critical period towards adulthood with many changes, including physical maturation, increased salience of peer interaction, and imbalance between emotional and cognitive systems [51,52,53,54,55]. Early adulthood (20–35 years old) progressively leads to the formation of a stable identity and cognitive maturity, including completion of education and moving into society [55,56]. Middle adulthood (35–55 years old) is a period of developing intact family and social life, being seen as a peak in competence and ability to handle stress [57,58,59]. People in old age (55+ years) are more concerned about emotions and meaning. Although the cognitive performance of older people decreases, older adults are more cooperative and generous than other age groups [60,61,62,63,64]. Thus, there should be age-related differences in anthropomorphism of nature, connectedness to nature, environmental guilt, and PEB.

According to these developmental characteristics among individuals, the present study presents four hypotheses related to these variables. First, anthropomorphism of nature may gradually decrease as people age. Compared to younger people, people in middle and old age may be less likely to anthropomorphize agents, because they possess more experience [65]. Second, connectedness to nature may weaken within mid–late adolescence, and rise within early adulthood. Because mid–late adolescence is considered as a “time-out” in nature–human relationships [66], and the level of connectedness to nature is low [67,68]. Third, environmental guilt may diminish with increasing age, because people have better developed emotion regulation strategies (e.g., suppressing emotion or appraising events in a positive manner) as they age [69,70]. We speculate that PEB increases with age based on prior studies [30,71,72,73]. In summary, we propose the following hypotheses:

**Hypothesis** **(H5a).**
*Anthropomorphism of nature decreases with age.*


**Hypothesis** **(H5b).**
*Connectedness to nature decreases with age in mid–late adolescence and gradually increases in early adulthood.*


**Hypothesis** **(H5c).**
*Environmental guilt decreases with age.*


**Hypothesis** **(H5d).**
*PEB increases with age.*


In conclusion, we posit that connectedness to nature and environmental guilt mediate the relationship between anthropomorphism of nature and PEB. Notably, it is currently less established whether these mediating effects fluctuate with age. In terms of the age-related differences in anthropomorphism of nature, connectedness to nature, environmental guilt, and PEB, it is necessary to conduct an exploratory investigation about the role of age in this mediating relationship. Thus, we will explore the age differences in the mediating effects of connectedness to nature and environmental guilt, in the relationship between anthropomorphism of nature and PEB.

## 3. Methods

### 3.1. Measures

As shown in Table 1, the measurement items for all constructs were adapted from prior studies. These measures included anthropomorphism of nature [32,74], connectedness to nature [37,75], environmental guilt [76,77,78,79], and PEB [74,80,81,82]. The demographic variables included age, gender, and residential area. All these items were rated with a 7-point Likert scale, ranging from 1 (completely disagree) to 7 (completely agree). In addition, the factor loadings of all items were greater than 0.4 [83], so that all items were retained. Since the participants were Chinese, the expressions were polished to ensure the accuracy of the questionnaire. First, we invited five experts to review and revise all the items. Second, to further enhance the validity [84], we performed a pilot test and randomly selected 30 respondents to review these items. Based on the pilot test, we modified the expressions of the items to avoid unnecessary bias, and finally formed the questionnaire for this study.

### 3.2. Sampling and Data Collection

We conducted a questionnaire to collect data. Adult participants were recruited online. The questionnaire link was distributed through a professional interview response platform and the participants were recruited to respond. Each participant would be rewarded an RMB of 3–5 yuan for completing the questionnaire. The region distribution of the respondents covered 30 provinces, autonomous regions, and municipalities in China. Adolescent participants were from four high schools in the northeastern region (Chaoyang, Shenyang in Liaoning Province), northern region (Beijing), and the southern region (Qingyuan in Guangdong Province) of China. According to the Chinese census data, Beijing, Liaoning, and Guangdong are the most populous regions [85]. Qingyuan, Chaoyang, and Shenyang are the top populous cities in the provinces of Guangdong and Liaoning. In total, 1616 people responded. The questionnaire included attention-check items to improve the data validity (e.g., “Please select ‘completely agree’ for this question”). After removing the data that did not pass these tests, a total of 1364 questionnaires were obtained, with 84.41% of effective response.

Table 2 shows the demographic characteristics of the participants. Out of 1364 participants, 24.48% were in mid–late adolescence (15–20 years old) [64]; 29.47% were in early adulthood (20–35 years old) [53]; 23.53% were in middle adulthood (35–55 years old) [56]; 22.51% were in old age (above 55 years old) [57,59,86]. With regards to gender, just over half of the participants were female (54.84%). Regarding the residential area of the participants, 52.64% lived in urban areas, and 47.36% lived in rural areas. Specifically, independent sample *t*-tests revealed that anthropomorphism of nature in female was significantly stronger than that in male (*p* = 0.01), indicating that gender could affect anthropomorphism of nature. We also found that anthropomorphism of nature in rural areas was significantly stronger than that in urban areas (*p* = 0.01). Therefore, we would consider gender and residential area as covariates for further analysis.

### 3.3. Analysis Method

We used SPSS 26.0, Origin 2021, and AMOS 23.0 to analyze the valid data. First, the reliability and validity were tested by Cronbach’s alpha, composite reliability (CR), average variance extracted (AVE), and confirmatory factor analysis (CFA). Second, we performed descriptive statistics and correlation analyses, and analyzed the life-span trajectories of the variables. Third, the structural equation model (SEM) was employed to analyze the direct effect between variables. Fourth, the bootstrap confidence interval method was used to test the mediating effect. Finally, a multi-group analysis was conducted for evaluation of the cross-age consistency of the mediating effect.

SEM can test and explain the relationships between these latent variables, when latent variables are contained in the model [87]. In addition, SEM can be used to test the fit of non-experimental data to the theoretical model [88]. SEM needs to estimate two models: the measurement model and the structural equation model [89]. The measurement model is used for evaluation of the relationships between the latent variables and observed variables, while the structural equation model is used for evaluation of the relationships among the latent variables.

## 4. Results

### 4.1. Reliability and Validity

First, we evaluated the questionnaire structure by testing Cronbach’s alpha. As shown in Table 1, the Cronbach’s alpha for anthropomorphism of nature, connectedness to nature, environmental guilt, and PEB were distributed between 0.773 and 0.954, indicating high reliability (greater than 0.7).

Second, composite reliability (CR) and average variance extracted (AVE) were used to measure the convergent validity. According to Fornell and Larcker [90], the convergent validity of the construct is acceptable if the CR and AVE values exceed 0.58 and 0.41, respectively. Hence, the construct reliability and convergent validity were ascertained (see Table 1).

Finally, we used CFA to test the fitness of the measurement model (including anthropomorphism of nature, connectedness to nature, environmental guilt, and PEB); χ^2^/*df* = 4.30, NFI (normed fit index) = 0.91, GFI (goodness of fit index) = 0.92, CFI (comparative fit index) = 0.93, RMSEA (root mean square error of approximation) = 0.05, and SRMR (standardized root mean square residual) = 0.05. The norm chi-square value χ^2^/*df* ranged between 2 and 5; NFI, GFI, and CFI were more than 0.90; RMSEA and SRMR were less than 0.08 and 0.06, respectively, being regarded as an acceptable model fit. These results indicated that the measurement model had a good fitting effect.

### 4.2. Correlation Analysis and Life-Span Trajectories of Variables

Table 3 shows the mean, standard deviation, and correlation coefficient of the variables in the total sample and in the four age groups. These results indicated that anthropomorphism of nature was positively correlated with connectedness to nature, environmental guilt, and PEB; connectedness to nature was positively correlated with environmental guilt and PEB; and environmental guilt was positively correlated with PEB (*p* < 0.01).

We further reported the correlation between each variable and age (see Table 3). Notably, age was modeled as a continuous variable and not as a categorical variable. In the total sample, anthropomorphism of nature and environmental guilt were negatively correlated with age; PEB was positively correlated with age. Connectedness to nature was negatively correlated with age in mid–late adolescence and positively correlated with age in early adulthood. The correlation results provided a basis for further analysis.

The present study further examined the trajectories of the four variables across the life span based on analysis from a prior study [91]. We fitted each of the measures on linear, quadratic, cubic, quartic, quintic, and sextic age and compared the R^2^. The analyses suggested sextic trajectories for the anthropomorphism of nature, connectedness to nature, and PEB, and a quintic trajectory for environmental guilt.

Figure 2 shows the predicted trajectories. The results showed that connectedness to nature decreased by approximately one-half standard deviation (*d* = −0.67) from mid–late adolescence to early adulthood, reached a minimum at about 20 years of age, and then increased by approximately one-third standard deviation (*d* = 0.29) from age 20 to 35 years. Environmental guilt decreased by approximately one-half standard deviation (*d* = −0.47) from mid–late adolescence to old age, reaching a minimum at about 76 years of age. The PEB increased by about one standard deviation (*d* = 1.15) from mid–late adolescence to old age, reaching a plateau at approximately 76 years of age. In contrast to the hypothesis H5a, anthropomorphism of nature did not decline with age. It declined by about one-half standard deviation (*d* = −0.52) from mid–late adolescence to early adulthood, reached a minimum at approximately 20 years of age, remained relatively stable until 50 years of age, and then rose by approximately one-half standard deviation (*d* = 0.65) from 50 to 76 years of age. Thus, H5b, H5c, and H5d were supported, but H5a was not supported.

### 4.3. Structural Model Analysis

#### 4.3.1. Direct Effect

We used a structural model to determine the significance of the hypothesis paths and the predictive capability of the model. The hypothesized model was estimated using bootstrapping procedure with 5000 samples. A *p*-value less than 0.05 indicates statistical significance. We further tested the fitness of the structural model, and the results indicated a good model fit (χ^2^/*df* = 4.03, NFI = 0.91, GFI = 0.92, CFI = 0.93, RMSEA = 0.05, SRMR = 0.04) that could be used to test the hypotheses.

The results are shown in Figure 3 and Table 4. Anthropomorphism of nature was positively and significantly associated with connectedness to nature (β = 0.358, *p* < 0.001) and environmental guilt (β = 0.095, *p* = 0.001). However, anthropomorphism of nature was not significantly related to PEB (β = 0.018, *p* = 0.551). Thus, H1 was not supported. However, connectedness to nature was positively and significantly associated with environmental guilt (β = 0.633, *p* < 0.001) and PEB (β = 0.236, *p* < 0.001). Notably, connectedness to nature had the strongest effect on environmental guilt. Environmental guilt was a positive and significant predictor of PEB (β = 0.236, *p* < 0.001). In conclusion, H4a was supported.

#### 4.3.2. Indirect Effect

Mediating-effect analysis is a commonly used statistical analysis method that examines how the independent variable affects the dependent variable through the mediating variables. Furthermore, SEM is a way to provide strong evidence for mediating effects [92]. Thus, the mediating effect was adopted to test the connection between anthropomorphism of nature and PEB via connectedness to nature and environmental guilt. As shown in Table 4, anthropomorphism of nature had a significant indirect effect on PEB via connectedness to nature (β = 0.023, *p* < 0.001) and environmental guilt (β = 0.005, *p* = 0.007), respectively.

In addition, connectedness to nature and environmental guilt had a serial mediating effect on anthropomorphism of nature and PEB (β = 0.011, *p* < 0.001). Further comparing the size of the indirect relationships, most of the indirect relationships accounted for the separate mediating effects of connectedness to nature and the serial mediating effect of connectedness to nature and environmental guilt. In short, the results of the mediating model indicated that connectedness to nature and environmental guilt were two key mediators between anthropomorphism of nature and PEB. Thus, H2, H3, and H4b were supported.

#### 4.3.3. Multi-Group Analysis

The effect of age on the mediation model was examined. First, we tried to include age in the mediation model (see Figure 4). However, the fit indicators with age were worse than those indicators without age (see Table 5). Therefore, a multi-group analysis was conducted to check whether the mediation model differed significantly among the four age groups. In other words, we explored whether there was cross-age stability of the mediation model.

The multi-group analysis needs to compare a constrained model that restricted the same paths among the four age groups and an unconstrained model with no restrictions. In general, the test needs to subtract the unconstrained model from the chi-square value of the constrained model [93]. If there is no significant difference in the chi-square value (*p* > 0.05), it can be assumed that the model has cross-age consistency. As the sample size easily affects the chi-square value, we needed to examine ∆CFI and ∆TLI (Tucker–Lewis index) to improve the judgment accuracy. If ∆CFI ≤ 0.01 or ∆TLI ≤ 0.02, it means that the model has multi-group consistency [94].

Our results indicated that the hypothesized structural model did not have significant differences in the chi-square values for age (*p* > 0.05), even though we took ΔCFI (<0.01) and ΔTLI (<0.01) into consideration. Thus, we concluded that the hypothesized structural model had invariance across different age groups.

## 5. Discussion

This study provides new insights from the perspective of unity between nature and humans. We investigated the underlying mechanisms that may account for the effect of anthropomorphism of nature on PEB. Anthropomorphism of nature could positively influence PEB through the mediating effects of connectedness to nature and environmental guilt. To our knowledge, these findings are the first to demonstrate the serial mediating roles of connectedness to nature and environmental guilt.

Since age is a crucial factor in PEB [10,28], we further considered age-related differences. Notably, the results revealed significant age-related changes: (1) anthropomorphism of nature decreased from mid–late adolescence to early adulthood, and increased in old age; (2) environmental guilt decreased with age; (3) PEB increased with age; (4) connectedness to nature decreased in mid–late adolescence and increased in early adulthood; and (5) the mediating effect had cross-age invariance. These findings enhance our understanding of age differences in anthropomorphism of nature, connectedness to nature, environmental guilt, PEB, and the mediating effect.

### 5.1. Relationships between Anthropomorphism of Nature, Connectedness to Nature, Environmental Guilt, and PEB

The present study investigated how anthropomorphism of nature affects PEB. The results showed that the relationship between anthropomorphism of nature and PEB was completely indirect. Specifically, anthropomorphism of nature could influence PEB through connectedness to nature and environmental guilt, respectively. We found that the mediating effect of environmental guilt was weaker than that of connectedness to nature. This highlights the key role of connectedness to nature in how anthropomorphism of nature induces environmental guilt. Our results also support the view that the link between connectedness to nature and environmental guilt is strong.

Based on this, a novel pathway for anthropomorphism of nature, positively affecting PEB, was found. In other words, connectedness to nature and environmental guilt were involved in serial mediation, wherein anthropomorphism of nature positively predicted connectedness to nature, which, in turn, positively predicted environmental guilt, and consequently PEB. Anthropomorphism of nature could allow people to establish and strengthen connections with nature [18]. By making people perceive the mind of nature, people’s self-representation may overlap with their representation of nature, further generating a sense of intimacy with nature. Based on this intimate relationship with nature, when people believe that their own or others’ actions have harmed nature, they are more likely to feel guilty. To reduce the negative experience of environmental guilt, people will further develop more PEBs to compensate for nature. In conclusion, anthropomorphism of nature, connectedness to nature, and environmental guilt all have profound impacts on promoting PEB.

### 5.2. Age-Related Differences

The present study took age into consideration, and most of our predictions regarding age were supported.

Surprisingly, anthropomorphism of nature did not diminish with age. It only diminished in mid–late adolescence to early adulthood, and increased again with old age. This result is inconsistent with that of previous studies at old age [65]. The reason may be the different agents of anthropomorphism used between the present study and the study reported by Letheren et al. [65]. Specifically, Letheren et al. [65] examined anthropomorphism of commercial products, while we examined anthropomorphism of nature. According to the three-factor theory of anthropomorphism [17], anthropomorphism is a reasoning process for the mind of nonhuman agents. The knowledge of human beings is the most accessible, because people have many experiences of being human. Therefore, people use their own experiences to infer the mental states of nonhuman agents. The anthropomorphizing process may be dominated by egocentricity. Once people acquire the requisite non-egocentric information, they will correct the egocentric simulation. Egocentricity is still detectable in mid–late adolescence, and diminishes in adulthood [61,95,96]. Thus, mid–late adolescents are more likely to anthropomorphize nonhuman agents due to egocentricity and a lack of alternative experience, both for commercial products and nature. As people mature, they acquire more experience and theories to understand nonhuman agents, thus people in early and middle adulthood are less likely to anthropomorphize.

However, in old age, the trajectories of anthropomorphism in nature and commercial products showed a difference. This may be driven by the elicited agent knowledge and effectance motivation. As people age, they will have a lot of opportunities to use their prior experiences. As a result, older adults are less likely to anthropomorphize commercial products. For anthropomorphism of nature, although older adults gain more knowledge about nature as they age, it is still not enough to have a comprehensive and deep understanding of nature. Relative to nature, humans are still too small. In addition, egocentricity appears again in old age [61]. Therefore, based on the need for a sense of understanding of nature and the appearance of egocentricity, older adults tend to infer more about nature with their own knowledge. Thus, older adults have a stronger tendency to anthropomorphize nature.

Connectedness to nature weakens with increasing age in mid–late adolescence, and remained stable after elevation in early adulthood. This result is consistent with findings reported by Hughes et al. [97], although the scales used are different. Hughes et al. [97] adopted the short-form Nature Relatedness Scale (NR-6) and the Connection to Nature Index (CNI). In particular, the NR-6 was used to measure cognitive, emotional and experiential connection to nature [98]. CNI was originally used to measure children’s trait nature connections [99]. The present study used the Connectedness to Nature Scale, which focuses on emotional connections to nature, and found the similar life-span trajectory. This finding is complementary to the trajectories of connectedness to nature measured by different scales. Our result further provides additional and strong evidence for age-related changes in connectedness to nature.

The self may be related to the changes in connectedness to nature. Interdependence with nature involves an extensive overlap representation between the self and nature. When people focus more on the self than on nature, their inclusion with nature will decrease. This leads to a lower level of connectedness to nature. Late adolescence is a period of self-focus. This period needs to focus on clarifying self-identity and solving the problem of “who am I” [100]. Consequently, adolescents are temporarily alienated from nature. Once these problems are resolved in early adulthood, the relationship with nature is restored gradually.

Environmental guilt diminished with increasing age. This result is consistent with the perspective that interpersonal guilt decreases with age [69,70]. This suggests that environmental guilt has a similar life-span trajectory to interpersonal guilt. Besides, our results support neuro-scientific research, as subcortical regions involved in emotion activation (e.g., amygdala, ventral striatum) tend to mature in mid–late adolescence, whereas the prefrontal lobes involved in emotion regulation are still developing and do not mature until early adulthood [52,101]. This maturation time may influence the development of environmental guilt. This means that adolescents are more likely to experience frequent and intense guilt. The experience of guilt may diminish, as the prefrontal cortex involved in emotion regulation matures in early adulthood [51]. Furthermore, as people age, they master more emotion-regulation skills [70,102]. Thus, older people may be less likely to feel guilty about the environmental degradation than younger people.

The incidence of PEB elevated with age, and this finding is consistent with some prior studies [68,71,72,73]. PEB occurs infrequently in mid–late adolescence. One possible reason is that adolescents lack environmental knowledge and do not perceive PEB as obligatory [68,103]. After acquiring more environmental knowledge and realizing the importance of protecting the environment in early and middle adulthood, PEB will increase. Additionally, older people show greater altruism [63], and they are more likely to protect the environment. Thus, PEB increases further in old age.

In addition, we examined whether there were differences in the serial mediation model across age groups. We demonstrated that the serial mediating effects of connectedness to nature and environmental guilt, in the relationship between anthropomorphism of nature and PEB, were stable across ages. It is evident that the serial mediating effect has general applicability in mid–late adolescence, early adulthood, middle adulthood, and old age. The study reveals that anthropomorphism of nature can bring different-aged people closer to nature, enhance their guilt toward nature, and promote conservation behaviors.

### 5.3. Implications

Our findings have substantial implications for interventions to promote PEB through anthropomorphism of nature. For the promotion of environmentally friendly behaviors, our work suggests that for different age groups, nature can be anthropomorphized to make people realize that nature has thoughts and feelings. Anthropomorphizing nature can allow people to connect emotionally with nature and establish a sense of oneness with nature. Based on this human–nature proximity, and because environmental guilt is a proximal variable affecting PEB, the effectiveness of the intervention may be further enhanced if environmental guilt is further encouraged. In details, when people realize that nature has consciousness, and know that human beings harm nature that has a mind, they would perform more PEBs.

Furthermore, our findings may have several implications in the context of the COVID-19 pandemic. The COVID-19 pandemic is uncontrollable for people and may reduce their sense of efficacy towards the environment [17]. Thus, in order to regain a sense of control, people’s tendency to anthropomorphize nature may be elevated and more likely to attribute human characteristics to nature [104]. In describing COVID-19, the General Secretary of the United Nations, António Guterres, said that “nature is sending us a clear message” and that “nature always strikes back and it is already doing so with growing force and fury” [105]. Therefore, the effectiveness of anthropomorphism of nature may be enhanced due to COVID-19. On the other hand, the COVID-19 pandemic has imposed significant restrictions on people’s travel. In order to prevent or slow down the spread of the virus, people had to stay at home. Thus, there were many opportunities for people to engage in PEBs in the private sphere, such as green purchasing, energy conservation, or recycling. And the focus of this study is also on the promotion of PEBs in the private sphere, through anthropomorphism of nature. In addition, to increase the impact of the green outreach worldwide, environmental appeals can also be made through the Belt and Road Initiative, with strategies that anthropomorphize nature.

### 5.4. Limitations and Future Directions

There are several limitations that can be addressed in future research. First, the present study adopted a self-report survey for data collection. Further studies are recommended to conduct experiments that manipulate anthropomorphism of nature and measure connectedness to nature, environmental guilt, and the actual PEBs. Second, age-related trajectories in anthropomorphism of nature, connectedness to nature, environmental guilt, and PEB were examined based on a cross-sectional study, and researchers can further conduct longitudinal studies. Researchers can track the same people and observe the trajectories of the variables above and the age differences in psychological mechanisms. Third, the present study focused on PEBs in the private sphere (e.g., green consumption, energy saving, and recycling). Further exploration of other types of PEBs in the public sphere (e.g., support environmental policies or join environmental organizations) is recommended. Fourth, the reliability coefficient for the “anthropomorphism of nature” scale in this study seems too high; some items are probably redundant. For future scale development, it is recommended to consider more heterogeneous items. Furthermore, there are other potential mediators which may account for the effect of anthropomorphism of nature on PEB, such as empathy with nature and gratitude towards nature. These two factors are significantly and positively associated with anthropomorphism of nature, connectedness to nature, and PEB [25,33]. However, these studies only considered the effects of empathy with nature and gratitude towards nature separately. In order to establish a more complete system of nature-related variables, future studies can simultaneously examine anthropomorphism of nature, connectedness to nature, environmental guilt, empathy with nature, and gratitude towards nature, exploring their relationships and their effects on pro-environmental behavior.

## 6. Conclusions

The present study indicates the underlying processes in the influence of anthropomorphism of nature on PEB, and reports these age-related changes (15–75 years). First, we find that anthropomorphism of nature positively affects PEB through the mediation of connectedness to nature and environmental guilt. The mediating effect shows cross-age stability. Second, as age increases, people perform behaviors in a more environmentally friendly or green way (more PEB), and feel less environmental guilt. Anthropomorphism of nature decreases in mid–late adolescence, and increases in old age. Late adolescence has a low level of connectedness to nature. Our findings show that in the field of pro-environmental behaviors, across different age groups, nature can be anthropomorphized to make people realize that nature is a system of conscious thought. These will make people establish a sense of oneness with nature. Meanwhile, we can emphasize that people harm nature, and this would induce a sense of environmental guilt, which can promote more environmentally friendly behaviors. In conclusion, the present work can inspire researchers and policymakers to look beyond the human-centric perspective and consider how people live in harmony with nature, and motivate them to live in an eco-friendly way.

## Figures and Tables

**Figure 1 ijerph-20-02393-f001:**
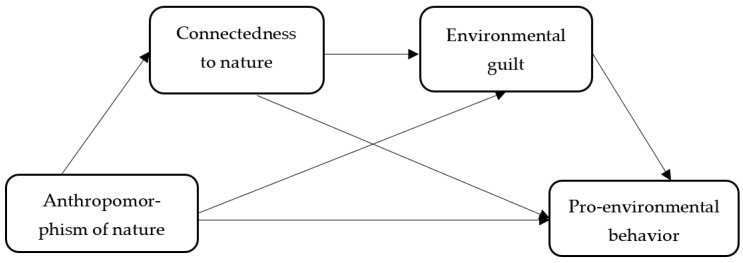
Conceptual model.

**Figure 2 ijerph-20-02393-f002:**
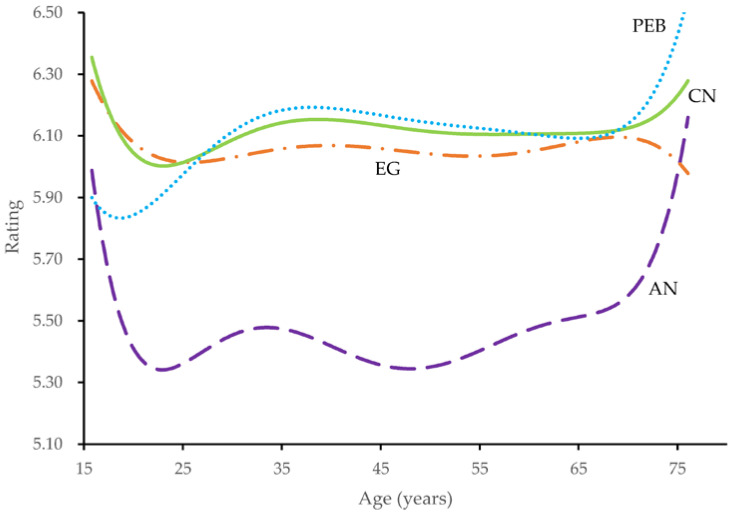
Trajectories of variables from 15 to 76 years of age. Note. PEB = Pro-environmental Behavior; CN = Connectedness to Nature; EG = Environmental Guilt; AN = Anthropomorphism of Nature.

**Figure 3 ijerph-20-02393-f003:**
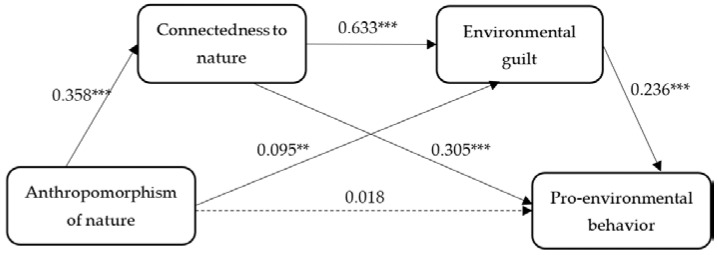
Structural model and standardized estimate values. Note. Solid arrows indicate statistically significant paths (** *p* < 0.01; *** *p* < 0.001), and dashed arrows indicate nonsignificant paths.

**Figure 4 ijerph-20-02393-f004:**
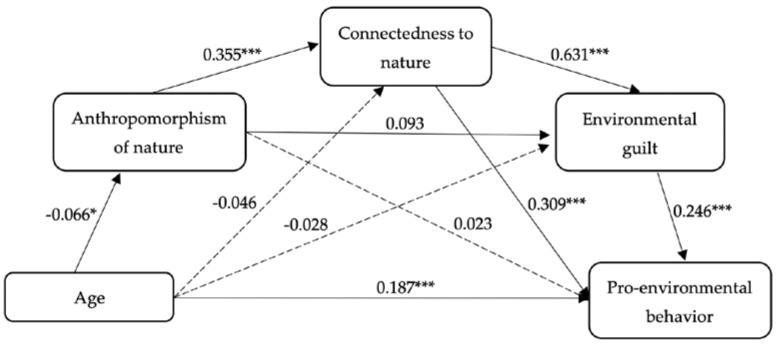
Structural model with age. Note. * *p* < 0.05; *** *p* < 0.001.

**Table 1 ijerph-20-02393-t001:** Measurement items.

Variables	Items	Loading	AVE	CR
Anthropomorphism of Nature (AN)			0.815	0.964
[32,74] α = 0.954	Nature has consciousness.	0.894		
	Nature has a mind of its own.	0.916		
	Ocean has its own free will.	0.901		
	Nature has intentions.	0.919		
	Forest has its own intention.	0.925		
	Nature has emotional experience.	0.858		
Connectedness to Nature (CN)			0.410	0.861
[37,75] α = 0.813	When I think of my life, I imagine myself to be part of a larger cyclical process of living.	0.639		
	I often feel a kinship with animals and plants.	0.554		
	Like a tree can be part of a forest, I feel embedded within the broader natural world.	0.707		
	I often feel part of the web of life.	0.677		
	I often feel a sense of oneness with the natural world around me.	0.655		
	I feel that all inhabitants of Earth, human, and nonhuman, share a common ‘life force’.	0.613		
	I feel as though I belong to the Earth as equally as it belongs to me.	0.574		
	I recognize and appreciate the intelligence of other living organisms.	0.637		
	I think of the natural world as a community to which I belong.	0.687		
Environmental Guilt (EG)			0.534	0.851
[76,77,78,79] α = 0.774	I feel bad when I think about human pollution damage toward nature.	0.666		
	I can easily feel guilty for the pollution damage humans caused to nature.	0.761		
	I feel regret for human pollution damage toward nature in the past.	0.781		
	I believe that I should repair the damage caused to nature by humans.	0.635		
	When I think about the pollution damage humans have caused to nature, I feel sorry.	0.798		
Pro-environmental Behavior (PEB)			0.503	0.888
[74,80,81,82] α = 0.770	I encourage others to sort.	0.752		
	I put electronic waste (e.g., used batteries, cell phones) into special recycling bins.	0.742		
	I sort the garbage and put it in corresponding bins (e.g., recyclable and non-recyclable).	0.764		
	I pick up litter when seeing it on the street, and put it in the bin.	0.701		
	I advise my family and friends to recycle or reuse products.	0.538		
	When the room is empty, I take the initiative to shut down the lights or air conditioning.	0.819		
	I use my own bag when shopping.	0.490		
	I turn off the tap when brushing teeth.	0.798		

Note. AVE = Average Variance Extracted; CR = Composite Reliability.

**Table 2 ijerph-20-02393-t002:** Demographic characteristics of respondents.

Variables	Category	Frequency	Percent
Age (years)	15–20	334	24.48
	20–35	402	29.47
	35–55	321	23.53
	>55	307	22.51
Gender	Male	616	45.16
	Female	748	54.84
Residential area	Urban	718	52.64
	Rural	646	47.36
Total		1364	100

**Table 3 ijerph-20-02393-t003:** Descriptive statistics and bivariate correlations of variables across different groups.

Variable	M ± SD	AN	CN	EG	PEB	Age
Total sample						
AN	5.50 ± 1.25	1				
CN	6.13 ± 0.53	0.33 **	1			
EG	6.08 ± 0.64	0.28 **	0.55 **	1		
PEB	6.06 ± 0.62	0.18 **	0.40 **	0.36 **	1	
Age (in years)	36.57 ± 15.80	−0.06 *	−0.05	−0.07 *	0.15 **	1
Mid–late adolescence						
AN	5.74 ± 1.31	1				
CN	6.22 ± 0.66	0.37 **	1			
EG	6.19 ± 0.76	0.27 **	0.57 **	1		
PEB	5.84 ± 0.76	0.22 **	0.45 **	0.41 **	1	
Age (in years)	17.26 ± 1.12	−0.07	−0.12 **	−0.09	−0.07	1
Early adulthood						
AN	5.44 ± 1.28	1				
CN	6.10 ± 0.48	0.24 **	1			
EG	6.05 ± 0.64	0.21 **	0.50 **	1		
PEB	6.09 ± 0.55	0.17 **	0.36 **	0.32 **	1	
Age (in years)	29.86 ± 3.97	0.07	0.13 *	0.03	0.07	1
Middle adulthood						
AN	5.37 ± 1.29	1				
CN	6.12 ± 0.48	0.38 **	1			
EG	6.05 ± 0.58	0.36 **	0.58 **	1		
PEB	6.19 ± 0.52	0.25 **	0.54 **	0.47 **	1	
Age (in years)	43.20 ± 5.85	−0.01	0.02	0.04	−0.04	1
Old age						
AN	5.47 ± 1.07	1				
CN	6.11 ± 0.48	0.34 **	1			
EG	6.04 ± 0.56	0.31 **	0.55 **	1		
PEB	6.11 ± 0.56	0.20 **	0.37 **	0.38 **	1	
Age (in years)	59.45 ± 4.16	0.04	0.02	0.03	0.02	1

Note. M = Mean; SD = Standard Deviation; AN = Anthropomorphism of Nature; CN = Connectedness to Nature; EG = Environmental Guilt; PEB = Pro-environmental Behavior; * *p* < 0.05, ** *p* < 0.01.

**Table 4 ijerph-20-02393-t004:** Summary of structural model results.

Path	Direct Effect		Indirect Effect	
	β	95% CI	β	95% CI
AN→PEB	0.018	[−0.056, 0.096]		
CN→PEB	0.305 ***	[0.199, 0.411]		
AN→CN	0.358 ***	[0.294, 0.422]		
EG→PEB	0.236 ***	[0.130, 0.340]		
AN→EG	0.095 **	[0.020, 0.174]		
CN→EG	0.633 ***	[0.552, 0.706]		
AN→CN→PEB			0.023 ***	[0.014, 0.035]
AN→EG→PEB			0.005 **	[0.001, 0.011]
AN→CN→EG→PEB			0.011 ***	[0.006, 0.019]

Note. AN = Anthropomorphism of Nature; PEB = Pro-environmental Behavior; CN = Connectedness to Nature; EG = Environmental Guilt; ** *p* < 0.01, *** *p* < 0.001.

**Table 5 ijerph-20-02393-t005:** Model comparison.

	χ^2^/*df*	CFI	TLI	RMSEA	SRMR
Model without age	4.03	0.93	0.92	0.05	0.05
Model with age	4.16	0.92	0.91	0.05	0.05

Note. CFI = Comparative Fit Index; TLI = Tucker-Lewis Index; RMSEA = Root Mean Square Error of Approximation; SRMR = Standardized Root Mean Square Residual.

## Data Availability

The data presented in this study are available on request from the corresponding author.

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
