# Peer review of "The Trajectory of Anthropomorphism and Pro-Environmental Behavior: A Serial Mediation Model"

_ijerph, 2023, doi:10.3390/ijerph20032393_

Round 1

Reviewer 1 Report

This is an interesting study, with an important topic. The theoretical rationale behind it is convincing, the methodology is solid, and the results are presented clearly.

The description of the sampling procedure is a bit vague: the authors state that 1,616 questionnaires were distributed. This implies that they were sent directly to potential respondents? How was the selection done, i.e. how did you target your participants? (e.g. did the agency target a stratified sample, reflecting the demographic structure within a given province, or was there some other criteria?) Or was it that the link was published, and then more general calls to participate were sent, and then 1,616 people responded? Please clarify this.

Furthermore, could you provide examples of tests to improve data validity? Those were some sort of attention check items, or what?

The reliability coefficient for the Anthropomorphism of Nature scale seems a bit too high. Some items are probably redundant. Consider including a more heterogeneous items in future developments of the scale.

The major issue I feel needs addressing in more detail: the authors described their finding nicely, and keep referring to the potential implications of these findings for future public campaigns. However, they do not provide any specific instructions for these campaigns. For example, how exactly should those who plan a campaign “encourage connectedness to nature and environmental guilt”?

Reviewer 2 Report

This research investigates the effect of anthropomorphism on pro-environmental behaviour directly, and indirectly via connectedness to nature and environmental guilt.

The chosen subject is valuable for the scholars and practitioners, however there are major issues need to be addressed to be warranted for publication:

1.      Last sentence of the abstract could include one important practical implications rather than generic statement.

2.      Definition of the anthropomorphism should be removed from the attract to the text.

3.      Last sentence of the introduction should highlight the contributions of the study as well as importance of it.

4.      Supportive arguments for H1 should be strengthened.

5.      In H2, there are two paths, one from the anthropomorphism to connected to nature, and another path is from connected to nature to PEB. The second part is not strong enough, and needs more justification. Same comment for H3,  which means, the path from environmental guilt to PEB should have been strengthened.. Also, for both hypothesis, the order of logical arguments should follow the same logic, first it should explain the first path, then the second one.

6.      H4a is redundant, as mentioned above H2 already covers this hypothesis. So H4 should only focus on serial mediator effect, that is how anthropomorphism affects connected to nature first, then how this variable affects environmental guilt, and how this leads to positive PEB.

7.      Age could be potential moderator on the model, right, then please show it on the Figure.

8.      In Table 1, could you please clarify whether some variables  that has lower factor loading were excluded from the analysis or not. This is also be shown in the Table.

9.      It would be better to categorise discussion and implications under Theoretical and Practical Implications.

10.  P.2, row 45 A should be a, no capital letter; p.3., row 137-38, Harth and colleagues should replaced with Harth et al. (). ; p.7, row 254 Of the participants, should replaced with Out of 1,364 participants?

Reviewer 3 Report

Recommendations: Major Revision

The paper's concept is good, and the manuscript tried to present a good idea to investigate the Trajectory of Anthropomorphism and Pro-environmental Behavior using a Serial Mediation Model. The present study uses sufficient datasets and tests the hypotheses based on a cross-sectional sample of 1,364 residents aged 15-76 years using structural equation modeling. Also, the results of the study may be an excellent addition to the general body of knowledge on the environment and health policy. The authors employ a relatively new approach to analyze the long-run evolution. However, the manuscript has the following shortcomings;

1.         The introduction is windy and has not the proper motivation for the study. There is a need to ascertain a clear novelty and relationship of variables of this paper.

2.         Although the sample is sufficient, and large dataset but still references are till 2021, which is not updated in time dimensions. Today end 2022, It’s better to have updated data considering post covid effects.

3.         The authors prove to know well the extant literature, but the text has to be reorganized.

4.         To bring more value to the paper, the author (s) should consider an extension of the literature that analyses the same relation but on similar countries as well. Literature review should define concept then theoretical background and finally linkage among variables and identify gap in current research and culminate in Conceptual Framework.  What has been done so far (combine related stuff), what left (literature gap), how this study is going to fill it.

5.         It would be better to see further the explanation of the consequences of the climate policy and regional especially BRI What kind of systematic and sustained policy intervention can be more productive, especially after Covid-19? Could you please brief more in recommendations.

6.         Although this is a good technique to test, however, the author still needs to present the results of the study clearly with proper justification of previous studies' findings. The clearly presented results may be a good addition to knowledge. 

7.         The reference style seems not appropriate. The citation style should be consistent throughout the manuscript. Also, the end reference style should be consistent

8.         The conclusion needs to be further elaborated. Recommendations are general in nature.

9.         What are the limitations of the study and areas of future research?- the authors are expected to talk about in greater detail.

10.     The concept under study is useful, and the author tried to present an addition in knowledge, but the manuscript needed a good motivation of the study along with justification why these techniques are adding value in practice. The manuscript requires a good flow and better presentation of ideas along with clear results.

It is recommended to revise the manuscript with consideration of the above-mentioned comments.
